# Certifying Confidence via Randomized Smoothing

**Aounon Kumar**
University of Maryland
aounon@umd.edu

**Alexander Levine**
University of Maryland
alevine0@cs.umd.edu

**Soheil Feizi**
University of Maryland
sfeizi@cs.umd.edu

**Tom Goldstein**
University of Maryland
tomg@cs.umd.edu

## Abstract

Randomized smoothing has been shown to provide good certified-robustness guarantees for high-dimensional classification problems. It uses the probabilities of predicting the top two most-likely classes around an input point under a smoothing distribution to generate a certified radius for a classifier's prediction. However, most smoothing methods do not give us any information about the *confidence* with which the underlying classifier (e.g., deep neural network) makes a prediction. In this work, we propose a method to generate certified radii for the prediction confidence of the smoothed classifier. We consider two notions for quantifying confidence: average prediction score of a class and the margin by which the average prediction score of one class exceeds that of another. We modify the Neyman-Pearson lemma (a key theorem in randomized smoothing) to design a procedure for computing the certified radius where the confidence is guaranteed to stay above a certain threshold. Our experimental results on CIFAR-10 and ImageNet datasets show that using information about the distribution of the confidence scores allows us to achieve a significantly better certified radius than ignoring it. Thus, we demonstrate that extra information about the base classifier at the input point can help improve certified guarantees for the smoothed classifier. Code for the experiments is available at https://github.com/aounon/cdf-smoothing.

## 1 Introduction

Deep neural networks have been shown to be vulnerable to adversarial attacks, in which a nearly imperceptible perturbation is added to an input image to completely alter the network's prediction [37, 31, 12, 21]. Several empirical defenses have been proposed over the years to produce classifiers that are robust to such perturbations [20, 4, 16, 8, 30, 14, 11]. However, without robustness guarantees, it is often the case that these defenses are broken by stronger attacks [5, 1, 40, 39]. Certified defenses, such as those based on convex-relaxation [41, 33, 35, 6, 36] and interval-bound propagation [13, 17, 9, 32], address this issue by producing robustness guarantees within a neighborhood of an input point. However, due to the complexity of present-day neural networks, these methods have seen limited use in high-dimensional datasets such as ImageNet.

Randomized smoothing has recently emerged as the state-of-the-art technique for certifying adversarial robustness with the scalability to handle datasets as large as ImageNet [22, 28, 7, 34]. This defense uses a base classifier, e.g. a deep neural network, to make predictions. Given an input image, a smoothing method queries the top class label at a large number of points in a Gaussian distribution surrounding the image, and returns the label with the majority vote. If the input image is perturbed slightly, the new voting population overlaps greatly with the smoothing distribution around the original image, and so the vote outcome can change only a small amount.

Conventional smoothing throws away a lot of information about class labels, and has limited capabilities that make its outputs difficult to use for decision making. Conventional classification networks with a softmax layer output a confidence score that can be interpreted as the degree of certainty the network has about the class label [15]. This is a crucial piece of information in real world decision-making applications such as self-driving cars [3] and disease-diagnosis networks [18], where safety is paramount.

In contrast, standard Gaussian smoothing methods take binary votes at each randomly sampled point – i.e., each point votes either for or against the most likely class, without conveying any information about how confident the network is in the class label. This may lead to scenarios where a point has a large certified radius but the underlying classifier has a low confidence score. For example, imagine a 2-way classifier for which a large portion, say 95%, of the sampled points predict the same class. In this case, the certified radius will be very large (indicating that this image is not an $\ell_2$-bounded adversarial example). However, it could be that each point predicts the top class with very low confidence. In this case, one should have very low confidence in the class label, despite the strength of the adversarial certificate. A Gaussian smoothing classifier counts a 51% confidence vote exactly the same way as a 99% confidence vote, and this important information is erased.

In this work, we restore confidence information in certified classifiers by proposing a method that produces class labels with a *certified confidence score*. Instead of taking a vote at each Gaussian sample around the input point, we average the confidence scores from the underlying base classifier for each class. The prediction of our smoothed classifier is given by the argmax of the expected scores of all the classes. Using the probability distribution of the confidence scores under the Gaussian, we produce a lower bound on how much the expected confidence score of the predicted class can be manipulated by a bounded perturbation to the input image. To do this, we adapt the Neyman-Pearson lemma, the fundamental theorem that characterizes the worst-case behaviour of the classifier under regular (binary) voting, to leverage the distributional information about the confidence scores. The lower bound we obtain is monotonically decreasing with the $\ell_2$-norm of the perturbation and can be expressed as a linear combination of the Gaussian CDF at different points. This allows us to design an efficient binary search based algorithm to compute the radius within which the expected score is guaranteed to be above a given threshold. Our method endows smoothed classifiers with the new and important capability of producing confidence scores.

We study two notions of measuring confidence: the *average prediction score* of a class, and the *margin* by which the average prediction score of one class exceeds that of another. The average prediction score is the expected value of the activations in the final softmax-layer of a neural network under the smoothing distribution. A class is guaranteed to be the predicted class if its average prediction score is greater than one half (since softmax values add up to one) or it maintains a positive margin over all the other classes. For both these measures, along with the bounds described in the previous paragraph, we also derive naive lower bounds on the expected score at a perturbed input point that do not use the distribution of the scores. We perform experiments on CIFAR-10 and ImageNet datasets which show that using information about the distribution of the scores allows us to achieve better certified guarantees than the naive method.

**Related work:** Randomized smoothing as a technique to design certifiably robust machine learning models has been studied amply in recent years. It has been used to produce certified robustness against additive threat models, such as, $\ell_1$ [22, 38], $\ell_2$ [7, 29, 27] and $\ell_0$-norm [23, 26] bounded adversaries, as well as non-additive threat models, such as, Wasserstein Adversarial attacks [24]. A derandomized version has been shown to provide robustness guarantees for patch attacks [25]. Smoothed classifiers that use the average confidence scores have been studied in [34] to achieve better certified robustness through adversarial training. A recent work uses the median score to generated certified robustness for regression models [43]. Differential privacy based defense method studied in [22] is capable of providing a guarantee on the test accuracy of a robust model under adversarial attack. Various limitations of randomized smoothing, like its inapplicability to high-dimensional problems for $\ell_\infty$-robustness, have been studied in [19, 42, 2].

## 2 Background and Notation

Gaussian smoothing, introduced by Cohen et al. in 2019, relies on a "base classifier," which is a mapping $f : \mathbb{R}^d \to \mathcal{Y}$ where $\mathbb{R}^d$ is the input space and $\mathcal{Y}$ is a set of $k$ classes. It defines a smoothed

classifier $\bar{f}$ as

$$\bar{f}(x) = \operatorname*{argmax}_{c \in \mathcal{Y}} \mathbb{P}(f(x + \delta) = c)$$

where $\delta \sim \mathcal{N}(0, \sigma^2 I)$ is sampled from an isometric Gaussian distribution with variance $\sigma^2$. It returns the class that is most likely to be sampled by the Gaussian distribution centered at point $x$. Let $p_1$ and $p_2$ be the probabilities of sampling the top two most likely classes. Then, $\bar{f}$ is guaranteed to be constant within an $\ell_2$-ball of radius

$$R = \frac{\sigma}{2} \left( \Phi^{-1}(p_1) - \Phi^{-1}(p_2) \right)$$

where $\Phi^{-1}$ is the inverse CDF of the standard Gaussian distribution [7]. For a practical certification algorithm, a lower bound $\underline{p_1}$ on $p_1$ and an upper bound $\overline{p_2} = 1 - \underline{p_1}$ on $p_2$, with probability $1 - \alpha$ for a given $\alpha \in (0, 1)$, are obtained and the certified radius is given by $R = \sigma \Phi^{-1}(\underline{p_1})$. This analysis is tight for $\ell_2$ perturbations; the bound is achieved by a worst-case classifier in which all the points in the top-class are restricted to a half-space separated by a hyperplane orthogonal to the direction of the perturbation.

In our discussion, we diverge from the standard notation described above, and assume that the base classifier $f$ maps points in $\mathbb{R}^d$ to a $k$-tuple of confidence scores. Thus, $f : \mathbb{R}^d \to (a, b)^k$ for some $a, b \in \mathbb{R}$ and $a < b$[1]. We define the smoothed version of the classifier as

$$\bar{f}(x) = \mathbb{E}_{\delta \sim \mathcal{N}(0, \sigma^2 I)} [f(x + \delta)],$$

which is the expectation of the class scores under the Gaussian distribution centered at $x$. The final prediction is made by taking an argmax of the expected scores. This definition has been studied by Salman et al. in [34] to develop an attack against smoothed classifiers which when used in an adversarial training setting helps boost the performance of conventional smoothing. The goal of this work is to identify a radius around an image $x$ within which the expected confidence score of the predicted class $i$, i.e. $\bar{f}_i(x) = \mathbb{E}[f_i(x + \delta)]$, remains above a given threshold $c \in (a, b)$[2].

We measure confidence using two different notions. The first measure is the average prediction score of a class as output by the final softmax layer. We denote the prediction score function with $h : \mathbb{R}^d \to (0, 1)^k$ and define the average for class $i$ as $\bar{h}_i(x) = \mathbb{E}[h_i(x + \delta)]$. The second one is the margin $m_i(x) = h_i(x) - \max_{j \neq i} h_j(x)$ by which class $i$ beats every other class in the softmax prediction score. In section 4, we show that the expected margin $\bar{m}_i(x) = \mathbb{E}[m_i(x + \delta)]$ for the predicted class is a lower-bound on the gap in average prediction scores of the top two class labels. Thus, $\bar{m}_i(x) > 0$ implies that $i$ is the predicted class.

## 3 Certifying Confidence Scores

Standard Gaussian smoothing for establishing certified class labels essentially works by averaging binary (0/1) votes from every image in a Gaussian cloud around the input image, $x$. It then establishes the worst-case class boundary given the recorded vote, and produces a certificate. The same machinery can be applied to produce a naive certificate for confidence score; rather than averaging binary votes, we simply average scores. We then produce the worst-case class distribution, in which each class lives in a separate half-space, and generate a certificate for this worst case.

However, the naive certificate described above throws away a lot of information. When continuous-values scores are recorded, we obtain not only the average score, but also the *distribution* of scores around the input point. By using this distributional information, we can potentially create a much stronger certificate.

To see why, consider the extreme case of a "flat" classifier function for which every sample in the Gaussian cloud around $x$ returns the same top-class prediction score of 0.55. In this case, the average score is 0.55 as well. For a function where the *distribution* of score votes is concentrated at 0.55 (or any other value great than $\frac{1}{2}$), the average score will always remain at 0.55 for *any* perturbation to $x$, thus yielding an infinite certified radius. However, when using the naive approach that throws away the distribution, the worst-case class boundary with average vote 0.55 is one with confidence

score 1.0 everywhere in a half-space occupying 0.55 probability, and 0.0 in a half-space with 0.45 probability. This worst-case, which uses only the average vote, produces a very small certified radius, in contrast to the infinite radius we could obtain from observing the distribution of votes.

Below, we first provide a simple bound that produces a certificate by averaging scores around the input image, and directly applying the framework from [7]. Then, we describe a more refined method that uses distributional information to obtain stronger bounds.

## 3.1 A baseline method using Gaussian means

In this section, we describe a method that uses only the average confidence over the Gaussian distribution surrounding $x$, and not the distribution of values, to bound how much the expected score can change when $x$ is perturbed with an $\ell_2$ radius of $R$ units. This is a straightforward extension of Cohen et al.'s [7] work to our framework. It shows that regardless the behaviour of the base classifier $f$, its smoothed version $\bar{f}$ changes slowly which is similar to the observation of bounded Lipschitz-ness made by Salman et al. in [34] (Lemma 2). The worst-case classifier in this case assumes value $a$ in one half space and $b$ in other, with a linear boundary between the two as illustrated in figure 1a. The following theorem formally states the bounds, the proof of which is deferred to the appendix[3].

**Theorem 1.** *Let $\underline{e_i}(x)$ and $\overline{e_i}(x)$ be a lower-bound and an upper-bound respectively on the expected score $\bar{f}_i(x)$ for class $i$ and, let $\underline{p_i}(x) = \frac{\underline{e_i}(x) - a}{b - a}$ and $\overline{p_i}(x) = \frac{\overline{e_i}(x) - a}{b - a}$. Then, for a perturbation $x'$ of the input $x$, such that, $\|x' - x\|_2 \leq R$,*

$$\bar{f}_i(x') \geq b\Phi_\sigma(\Phi_\sigma^{-1}(\underline{p_i}(x)) - R) + a(1 - \Phi_\sigma(\Phi_\sigma^{-1}(\underline{p_i}(x)) - R)) \tag{1}$$

*and*

$$\bar{f}_i(x') \leq b\Phi_\sigma(\Phi_\sigma^{-1}(\overline{p_i}(x)) + R) + a(1 - \Phi_\sigma(\Phi_\sigma^{-1}(\overline{p_i}(x)) + R))$$

*where $\Phi_\sigma$ is the CDF of the univariate Gaussian distribution with $\sigma^2$ variance, i.e., $\mathcal{N}(0, \sigma^2)$.*

## 3.2 Proposed certificate

The bounds in section 3.1 are a simple application of the Neyman-Pearson lemma to our framework. But this method discards a lot of information about how the class scores are distributed in the Gaussian around the input point. Rather than consolidating the confidence scores from the samples into an expectation, we propose a method that uses the cumulative distribution function of the confidence scores to obtain improved bounds on the expected class scores.

Given an input $x$, we draw $m$ samples from the Gaussian distribution around $x$. We use the prediction of the base classifier $f$ on these points to generate bounds on the distribution function of the scores for the predicted class. These bounds, in turn, allow us to bound the amount by which the expected score of the class will decrease under an $\ell_2$ perturbation. Finally, we apply binary search to compute the radius for which this lower bound on the expected score remains above $c$.

Consider the sampling of scores around an image $x$ using a Gaussian distribution. Let the probability with which the score of class $i$ is above $s$ be

$$p_{i,s}(x) = \mathbb{P}_{\delta \sim \mathcal{N}(0, \sigma^2 I)}(f_i(x + \delta) \geq s).$$

For point $x$ and class $i$, consider the random variable $Z = -f_i(x + \delta)$ where $\delta \sim \mathcal{N}(0, \sigma^2 I)$. Let $F(s) = \mathbb{P}(Z \leq s)$ be the cumulative distribution function of $Z$ and $F_m(s) = \frac{1}{m} \sum_{j=1}^{m} \mathbf{1}\{Z_j \leq s\}$ be its empirical estimate. For a given $\alpha \in (0, 1)$, the Dvoretzky–Kiefer–Wolfowitz inequality [10] says that, with probability $1 - \alpha$, the true CDF is bounded by the empirical CDF as follows:

$$F_m(s) - \epsilon \leq F(s) \leq F_m(s) + \epsilon, \forall s,$$

where $\epsilon = \sqrt{\frac{\ln 2/\alpha}{2m}}$. Thus, $p_{i,s}(x)$ is also bounded within $\pm \epsilon$ of its empirical estimate $\sum_{j=1}^{m} \mathbf{1}\{f_i(x + \delta_j) \geq s\}$.

The following theorem bounds the expected class score under an $\ell_2$ perturbation using bounds on the cumulative distribution of the scores.

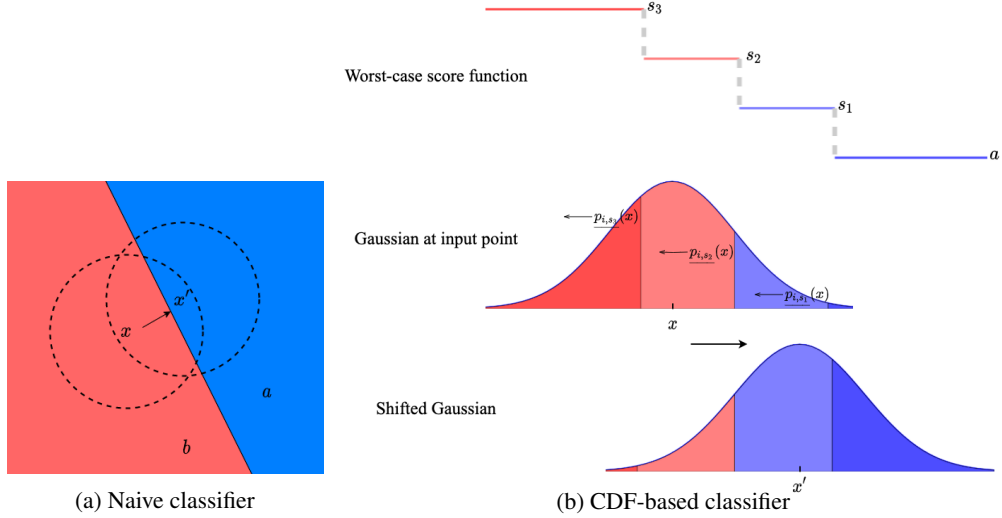

|(a) Naive classifier|(b) CDF-based classifier|

Figure 1: Worst case classifier behaviour using (a) naive approach and (b) CDF-based method. As the center of the distribution moves from $x$ to $x'$, the probability mass of the higher values of the score function (indicated in red) decreases and that of the lower values (indicated in blue) increases, bringing down the value of the expected score.

**Theorem 2.** *Let, for class $i$, $a < s_1 \leq s_2 \leq \cdots \leq s_n < b$ be $n$ real numbers and let $\overline{p_{i,s_j}}(x)$ and $\underline{p_{i,s_j}}(x)$ be upper and lower bounds on $p_{i,s_j}(x)$ respectively derived using the Dvoretzky–Kiefer–Wolfowitz inequality, with probability $1 - \alpha$, for a given $\alpha \in (0,1)$. Then, for a perturbation $x'$ of the input $x$, such that, $\|x' - x\|_2 \leq R$,*

$$\bar{f}_i(x') \geq a + (s_1 - a)\Phi_\sigma(\Phi_\sigma^{-1}(\underline{p_{i,s_1}}(x)) - R) + \sum_{j=2}^{n}(s_j - s_{j-1})\Phi_\sigma(\Phi_\sigma^{-1}(\underline{p_{i,s_j}}(x)) - R) \quad (2)$$

*and*

$$\bar{f}_i(x') \leq s_1 + (b - s_n)\Phi_\sigma(\Phi_\sigma^{-1}(\overline{p_{i,s_n}}(x)) + R) + \sum_{j=1}^{n-1}(s_{j+1} - s_j)\Phi_\sigma(\Phi_\sigma^{-1}(\overline{p_{i,s_j}}(x)) + R)$$

*where $\Phi_\sigma$ is the CDF of the univariate Gaussian distribution with $\sigma^2$ variance, i.e., $\mathcal{N}(0, \sigma^2)$.*

The above bounds are tight for $\ell_2$ perturbations. The worst-case classifier for the lower bound is one in which the class score decreases from $b$ to $a$ in steps, taking values $s_n, s_{n-1}, \ldots, s_1$ at each level. Figure 1b illustrates this case for three intermediate levels. A similar worst-case scenario can be constructed for the upper bound as well where the class score increases from $a$ to $b$ along the direction of the perturbation. Even though our theoretical results allow us to derive both upper and lower bounds for the expected scores, we restrict ourselves to the lower bound in our experimental results. We provide a proof sketch for this theorem in section 3.3. Our experimental results show that the CDF-based approach beats the naive bounds in practice by a significant margin, showing that having more information about the classifier at the input point can help achieve better guarantees.

**Computing the certified radius** Both the bounds in theorem 2 monotonic in $R$. So, in order to find a certified radius, up to a precision $\tau$, such that the lower (upper) bound is above (below) a certain threshold we can apply binary search which will require at most $O(\log(1/\tau))$ evaluations of the bound.

### 3.3 Proof of Theorem 2

We present a brief proof for theorem 2. We use a slightly modified version of the Neyman-Pearson lemma (stated in [7]) which we prove in the appendix.

**Lemma 3** (Neyman & Pearson, 1933). *Let $X$ and $Y$ be random variables in $\mathbb{R}^d$ with densities $\mu_X$ and $\mu_Y$. Let $h : \mathbb{R}^d \to (a, b)$ be a function. Then:*

1. *If $S = \left\{ z \in \mathbb{R}^d \mid \frac{\mu_Y(z)}{\mu_X(z)} \le t \right\}$ for some $t > 0$ and $\mathbb{P}(h(X) \ge s) \ge \mathbb{P}(X \in S)$, then $\mathbb{P}(h(Y) \ge s) \ge \mathbb{P}(Y \in S)$.*

2. *If $S = \left\{ z \in \mathbb{R}^d \mid \frac{\mu_Y(z)}{\mu_X(z)} \ge t \right\}$ for some $t > 0$ and $\mathbb{P}(h(X) \ge s) \le \mathbb{P}(X \in S)$, then $\mathbb{P}(h(Y) \ge s) \le \mathbb{P}(Y \in S)$.*

Set $X$ to be the smoothing distribution at an input point $x$ and $Y$ to be that at $x+\epsilon$ for some perturbation vector $\epsilon$. For a class $i$, define sets $\underline{S}_{i,j} = \{z \in \mathbb{R}^d \mid \mu_Y(z)/\mu_X(z) \le t_{i,j}\}$ for some $t_{i,j} > 0$, such that, $\mathbb{P}(X \in \underline{S}_{i,j}) = \underline{p_{i,s_j}}(x)$. Similarly, define sets $\overline{S}_{i,j} = \{z \in \mathbb{R}^d \mid \mu_Y(z)/\mu_X(z) \ge t'_{i,j}\}$ for some $t'_{i,j} > 0$, such that, $\mathbb{P}(X \in \overline{S}_{i,j}) = \overline{p_{i,s_j}}(x)$. Since, $\mathbb{P}(f_i(X) \ge s_j) \ge \mathbb{P}(X \in \underline{S}_{i,j})$, using lemma 3 we can say that $\mathbb{P}(f_i(Y) \ge s_i) \ge \mathbb{P}(Y \in \underline{S}_{i,j})$. Therefore,

$$
\begin{aligned}
\mathbb{E}[f_i(Y)] &\ge s_n \mathbb{P}(f_i(Y) \ge s_n) + s_{n-1}(\mathbb{P}(f_i(Y) \ge s_{n-1}) - \mathbb{P}(f_i(Y) \ge s_n)) \\
&\quad + \cdots + s_1(\mathbb{P}(f_i(Y) \ge s_1) - \mathbb{P}(f_i(Y) \ge s_2)) + a(1 - \mathbb{P}(f_i(Y) \ge s_1)) \\
&= a + (s_1 - a)\mathbb{P}(f_i(Y) \ge s_1) + \sum_{j=2}^{n} (s_j - s_{j-1})\mathbb{P}(f_i(Y) \ge s_j) \\
&\ge a + (s_1 - a)\mathbb{P}(Y \in \underline{S}_{i,1}) + \sum_{j=2}^{n} (s_j - s_{j-1})\mathbb{P}(Y \in \underline{S}_{i,j}).
\end{aligned}
$$

Similarly, $\mathbb{P}(f_i(X) \ge s_j) \le \mathbb{P}(X \in \overline{S}_{i,j})$ implies $\mathbb{P}(f_i(Y) \ge s_j) \le \mathbb{P}(Y \in \overline{S}_{i,j})$ as per lemma 3. Therefore,

$$
\begin{aligned}
\mathbb{E}[f_i(Y)] &\le b\mathbb{P}(f_i(Y) \ge s_n) + s_n(\mathbb{P}(f_i(Y) \ge s_{n-1}) - \mathbb{P}(f_i(Y) \ge s_n)) \\
&\quad + \cdots + s_1(1 - \mathbb{P}(f_i(Y) \ge s_1)) \\
&= (b - s_n)\mathbb{P}(f_i(Y) \ge s_n) + \sum_{j=1}^{n-1} (s_{j+1} - s_j)\mathbb{P}(f_i(Y) \ge s_j) + s_1 \\
&\le s_1 + (b - s_n)\mathbb{P}(Y \in \overline{S}_{i,n}) + \sum_{j=1}^{n-1} (s_{j+1} - s_j)\mathbb{P}(Y \in \overline{S}_{i,j}).
\end{aligned}
$$

Since, we are smoothing using an isometric Gaussian distribution with $\sigma^2$ variance, $\mu_X = \mathcal{N}(x, \sigma^2 I)$ and $\mu_Y = \mathcal{N}(x + \epsilon, \sigma^2 I)$. Then, for some $t$ and $\beta$

$$
\frac{\mu_Y(z)}{\mu_Y(z)} \le t \iff \epsilon^T z \le \beta
$$

$$
\frac{\mu_Y(z)}{\mu_Y(z)} \ge t \iff \epsilon^T z \ge \beta.
$$

Thus, each of the sets $\underline{S}_{i,j}$ and $\overline{S}_{i,j}$ is a half space defined by a hyper-plane orthogonal to the direction of the perturbation. This simplifies our analysis to one dimension, namely, the one along the perturbation. For each of the sets $\underline{S}_{i,j}$ and $\overline{S}_{i,j}$, we can find a point on the real number line $\Phi_\sigma^{-1}(\underline{p_{i,s_j}}(x))$ and $\Phi_\sigma^{-1}(\overline{p_{i,s_j}}(x))$ respectively such that the probability of a Gaussian sample to fall in that set is equal to the Gaussian CDF at that point. Therefore,

$$
\bar{f}_i(x + \epsilon) \ge a + (s_1 - a)\Phi_\sigma(\Phi_\sigma^{-1}(\underline{p_{i,s_1}}(x)) - R) + \sum_{j=2}^{n} (s_j - s_{j-1})\Phi_\sigma(\Phi_\sigma^{-1}(\underline{p_{i,s_j}}(x)) - R)
$$

and

$$
\bar{f}_i(x + \epsilon) \le s_1 + (b - s_n)\Phi_\sigma(\Phi_\sigma^{-1}(\overline{p_{i,s_n}}(x)) + R) + \sum_{j=1}^{n-1} (s_{j+1} - s_j)\Phi_\sigma(\Phi_\sigma^{-1}(\overline{p_{i,s_j}}(x)) + R)
$$

which completes the proof of theorem 2. We would like to note here that although we use the Gaussian distribution for smoothing, the modified Neyman-Pearson lemma does not make any assumptions on the shape of the distributions which allows for this proof to be adapted for other smoothing distributions as well.

# 4 Confidence measures

We study two notions of confidence: average prediction score of a class and the margin of average prediction score between two classes. Usually, neural networks make their predictions by outputting a prediction score for each class and then taking the argmax of the scores. Let $h : \mathbb{R}^d \to (0,1)^k$ be a classifier mapping input points to prediction scores between 0 and 1 for each class. We assume that the scores are generated by a softmax-like layer, i.e., $0 < h_i(x) < 1, \forall i \in \{1, \ldots, k\}$ and $\sum_i h_i(x) = 1$. For $\delta \sim \mathcal{N}(0, \sigma^2 I)$, we define average prediction score for a class $i$ as

$$\bar{h}_i(x) = \mathbb{E}[h_i(x + \delta)].$$

The final prediction for the smoothed classifier is made by taking an argmax over the average prediction scores of all the classes, i.e., $\mathrm{argmax}_i \ \bar{h}_i(x)$. Thus, if for a class $j$, $\bar{h}_j(x) \geq 0.5$, then $j = \mathrm{argmax}_i \ \bar{h}_i(x)$.

Now, we define margin $m$ at point $x$ for a class $i$ as

$$m_i(x) = h_i(x) - \max_{j \neq i} h_j(x).$$

Thus, if $i$ is the class with the highest prediction score, $m_i(x)$ is the lead it has over the second highest class (figure 2). And, for any other class $m_i(x)$ is the negative of the difference of the scores of that class with the highest class. We define average margin at point $x$ under smoothing distribution $\mathcal{P}$ as

$$\bar{m}_i(x) = \mathbb{E}[m_i(x + \delta)].$$

For a pair of classes $i$ and $j$, we have,

$$\begin{aligned} \bar{h}_i(x) - \bar{h}_j(x) &= \mathbb{E}[h_i(x + \delta)] - \mathbb{E}[h_j(x + \delta)] \\ &= \mathbb{E}[h_i(x + \delta) - h_j(x + \delta)] \\ &\geq \mathbb{E}[h_i(x + \delta) - \max_{j \neq i} h_j(x + \delta)] \\ &= \mathbb{E}[m_i(x + \delta)] = \bar{m}_i(x) \\ \bar{h}_i(x) &\geq \bar{h}_j(x) + \bar{m}_i(x). \end{aligned}$$

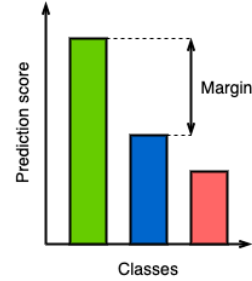

Figure 2: Margin

Thus, if $\bar{m}_i(x) > 0$, then class $i$ must have the highest average prediction score making it the predicted class under this notion of confidence measure.

# 5 Experiments

We conduct several experiments to motivate the use of certified confidence, and to validate the effectiveness of our proposed CDF-based certificate.

## 5.1 Does certified radius correlate with confidence score?

A classifier can fail because of an adversarial attack, or because of epistemic uncertainty – a class label may be uncertain or wrong because of lack of useful features, or because the model was not trained on sufficient representative data. The use of certified confidence is motivated by the observation that the original Gaussian averaging, which certifies the *security* of class labels, does not convey whether the user should be *confident* in the label because it neglects epistemic uncertainty. We demonstrate this with a simple experiment. In figure 3, we show plots of softmax prediction score vs. certified radius obtained using smoothed ResNet-110 and ResNet-50 classifiers trained by Cohen et al. in [7] for CIFAR-10 and ImageNet respectively. The noise level $\sigma$ used for this experiment was 0.25. For both models, the certified radii correlate very little with the prediction scores for the input images. The CIFAR-10 plot has points with high scores but small radii. While, for ImageNet, we see a lot of points with low scores but high radii. This motivates the need for certifying confidence; high radius does not imply high confidence of the underlying classifier. This lack of correlation is visualized in figure 4.

| Radius< 0.2, Confidence Score> 0.7 | Radius> 0.5, Confidence Score< 0.4 |

CIFAR-10 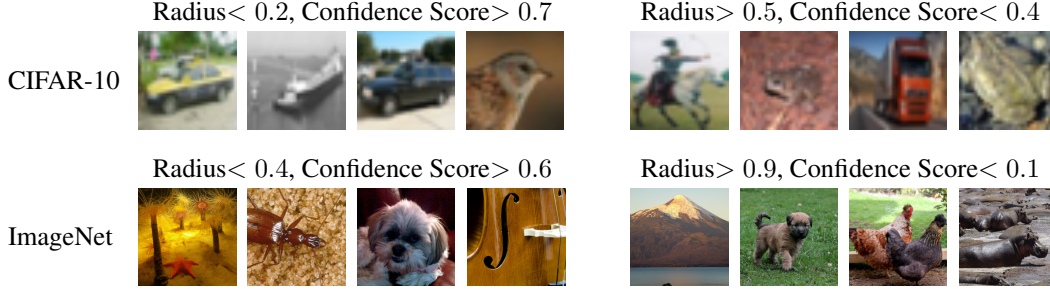

| Radius< 0.4, Confidence Score> 0.6 | Radius> 0.9, Confidence Score< 0.1 |

ImageNet

Figure 4: Certified radius does not correlate well with human visual confidence or network confidence score. Low radius images on the left have high confidence scores, while the high radius images on the right all have low confidence scores. There is not a pronounced visual difference between low- and high-radius images.

In the plots, CIFAR-10 images tend to have a higher prediction score than ImageNet images which is potentially due to the fact that the ImageNet dataset has a lot more classes than the CIFAR-10 dataset, driving the softmax scores down. There is a hard limit ($\sim 0.95$ for ImageNet) on the largest radius that can be generated by Cohen et al.'s certifying algorithm which causes a lot of the ImageNet points to accumulate at this radius value. This limit comes from the fact that even if all the samples around an input image vote for the same class, the lower-bound on the top-class probability is strictly less than one, which keeps the certified radius within a finite value.

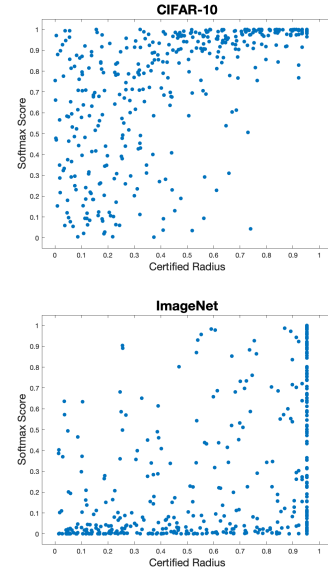

Figure 3: Prediction Score vs. Certified Radius.

## 5.2 Evaluating the strength of bounds

We use the ResNet-110 and ResNet-50 models trained by Cohen et al. in [7] on CIFAR-10 and ImageNet datasets respectively to generate confidence certificates. These models have been pre-trained with varying Gaussian noise level $\sigma$ in the training data. We use the same $\sigma$ for certifying confidences as well. We use the same number of samples $m = 100,000$ and value of $\alpha = 0.001$ as in [7]. We set $s_1, s_2, \ldots, s_n$ in theorem 2 such that the number of confidence score values falling in each of the intervals $(a, s_1), (s_1, s_2), \ldots, (s_n, b)$ is the same. We sort the scores from the $m$ samples in increasing order and set $s_i$ to be the element at position $1 + (i - 1)m/n$ in the order. We chose this method of splitting the range $(a, b)$, instead of at regular steps, to keep the intervals well-balanced. We present results for both notions of confidence measure: average prediction score and margin. Figure 5 plots certified accuracy, using the naive bound and the CDF-based method, for different threshold values for the top-class average prediction score and the margin at various radii for $\sigma = 0.25$. The same experiments for $\sigma = 0.50$ have been included in the appendix.

Each line is for a given threshold for the confidence score. The solid lines represent certificates derived using the CDF bound and the dashed lines are for ones using the naive bound. For the baseline certificate (1), we use Hoeffding's inequality to get a lower-bound on the expected top-class confidence score $e_i(x)$, that holds with probability $1 - \alpha$, for a given $\alpha \in (0, 1)$.

$$\underline{e_i}(x) = \frac{1}{m} \sum_{j=1}^{m} f_i(x + \delta_j) - (b - a)\sqrt{\frac{\ln(1/\alpha)}{2m}}$$

This bound is a reasonable choice because $\underline{p_i}(x)$ differs from the empirical estimate by the same amount $\sqrt{\ln(1/\alpha)/2m}$ as $\underline{p_{i,s}}(x)$ in the proposed CDF-based certificate. In the appendix, we also show that the baseline certificate, even with the best-possible lower-bound for $e_i(x)$, cannot beat our method for most cases.

We see a significant improvement in certified accuracy (e.g. at radius = 0.25) when certification is done using the CDF method instead of the naive bound. The confidence measure based on the margin

between average prediction scores yields slightly better certified accuracy when thresholded at zero than the other measure.

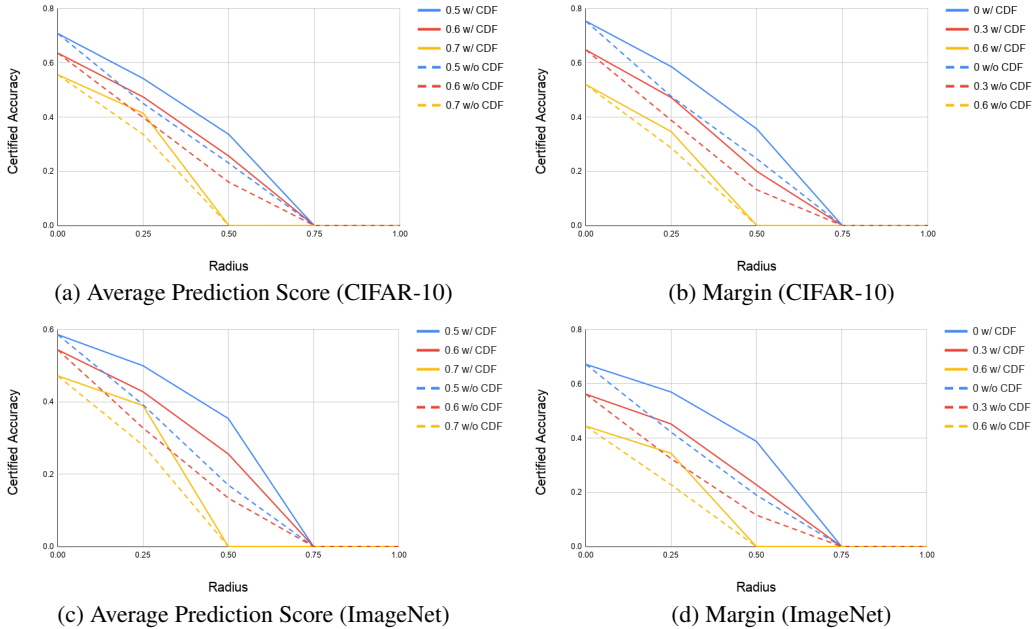

(a) Average Prediction Score (CIFAR-10)

(b) Margin (CIFAR-10)

(c) Average Prediction Score (ImageNet)

(d) Margin (ImageNet)

Figure 5: Certified accuracy vs. radius (CIFAR-10 & ImageNet) at different cutoffs for average confidence score with $\sigma = 0.25$. Solid and dashed lines represent certificates computed with and without CDF bound respectively.

# 6  Conclusion

While standard certificates can guarantee that a decision is *secure*, they contain little information about how *confident* the user should be in the assigned label. We present a method that certifies the confidence scores, rather than the labels, of images. By leveraging information about the distribution of confidence scores around an input image, we produce certificates that beat a naive bound based on a direct application of the Neyman-Pearson lemma. The results in this work show that certificates can be strengthened by incorporating more information into the worst-case bound than just the average vote. We hope this line of research leads to methods for strengthening smoothing certificates based on other information sources, such as properties of the base classifier or the spatial distribution of votes.

# 7  Broader Impact

We design procedures that equip randomized smoothing with certified prediction confidence, an important property for any real-world decision-making system to have. In applications where robustness is key, like credit scoring and disease diagnosis systems, it is important to know how certain the prediction model is about the output, so that a human expert can take over if the model's confidence is low.

However, this method does not produce any guarantees on the calibration of the underlying model itself. It could happen that the confidence measure used to determine the degree of certainty of the model does not actually reflect the probability of the prediction being correct. In other words, our methods depends on the underlying classifier to have high accuracy to perform reliably. With certificates, there is always a risk of conveying a false sense of confidence, but hopefully by producing interpretable risk scores along with certificates our work will help mitigate this problem.

## 8 Acknowledgments

We would like to thank anonymous reviewers for their valuable comments and suggestions. This work was supported by DARPA GARD, DARPA QED4RML, and the National Science Foundation Directorate of Mathematical Sciences. This project was supported in part by NSF CAREER AWARD 1942230, grants from NIST award 60NANB20D134, HR00112090132 and Simons Fellowship on "Foundations of Deep Learning."

## Footnotes

[1] $(a, b)$ denotes the open interval between $a$ and $b$.

[2] $f_i(x)$ denotes the $i$th component of $f(x)$

[3]A separate proof, using Lemma 2 from Salman et al. in [34], for this theorem for $\sigma = 1$ is also included in the appendix.

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
