[Supplementary Material · appendix_certifying_confidence.pdf]

# Appendix
# Certifying Confidence via Randomized Smoothing

**Aounon Kumar**
University of Maryland
aounon@umd.edu

**Alexander Levine**
University of Maryland
alevine0@cs.umd.edu

**Soheil Feizi**
University of Maryland
sfeizi@cs.umd.edu

**Tom Goldstein**
University of Maryland
tomg@cs.umd.edu

## A   Proof of Theorem 1

We first prove a slightly modified version of the Neyman-Pearson lemma.

**Lemma 1** (Neyman & Pearson, 1933). *Let $X$ and $Y$ be random variables in $\mathbb{R}^d$ with densities $\mu_X$ and $\mu_Y$. Let $h : \mathbb{R}^d \to (a, b)$ be a function. Then:*

*1. If $S = \left\{ z \in \mathbb{R}^d \mid \frac{\mu_Y(z)}{\mu_X(z)} \leq t \right\}$ for some $t > 0$ and $\mathbb{E}[h(X)] \geq (b-a)\mathbb{P}(X \in S) + a$, then $\mathbb{E}[h(Y)] \geq (b-a)\mathbb{P}(Y \in S) + a$.*

*2. If $S = \left\{ z \in \mathbb{R}^d \mid \frac{\mu_Y(z)}{\mu_X(z)} \geq t \right\}$ for some $t > 0$ and $\mathbb{E}[h(X)] \leq (b-a)\mathbb{P}(X \in S) + a$, then $\mathbb{E}[h(Y)] \leq (b-a)\mathbb{P}(Y \in S) + a$.*

*Proof.* Let $S^c$ be the complement set of $S$.

$$
\begin{aligned}
\mathbb{E}[h(Y)] - (b-a)\mathbb{P}(Y \in S) - a &= \mathbb{E}[h(Y)] - b\mathbb{P}(Y \in S) - a(1 - \mathbb{P}(Y \in S)) \\
&= \mathbb{E}[h(Y)] - b\mathbb{P}(Y \in S) - a\mathbb{P}(Y \notin S) \\
&= \int_{\mathbb{R}^d} h(z)\mu_Y(z)dz - b\int_S \mu_Y(z)dz - a\int_{S^c} \mu_Y(z)dz \\
&= \left[ \int_{S^c} h(z)\mu_Y(z)dz + \int_S h(z)\mu_Y(z)dz \right] - b\int_S \mu_Y(z)dz - a\int_{S^c} \mu_Y(z)dz \\
&= \int_{S^c} (h(z) - a)\mu_Y(z)dz - \int_S (b - h(z))\mu_Y(z)dz \\
&\geq t\left[ \int_{S^c} (h(z) - a)\mu_X(z)dz - \int_S (b - h(z))\mu_X(z)dz \right] \\
&\qquad\qquad\qquad\qquad\qquad\qquad\qquad\qquad \text{(since } a < h(z) < b\text{)} \\
&= t\left[ \int_{\mathbb{R}^d} h(z)\mu_X(z)dz - b\int_S \mu_X(z)dz - a\int_{S^c} \mu_X(z)dz \right] \\
&= t\left[ \mathbb{E}[h(X)] - b\mathbb{P}(X \in S) - a\mathbb{P}(X \notin S) \right] \\
&= t\left[ \mathbb{E}[h(X)] - b\mathbb{P}(X \in S) - a(1 - \mathbb{P}(X \in S)) \right] \\
&= t\left[ \mathbb{E}[h(X)] - (b-a)\mathbb{P}(X \in S) - a \right] \geq 0
\end{aligned}
$$

The second statement can be proven similarly by switching $\geq$ and $\leq$. $\qquad\square$

In the first statement of the lemma, set $h$ to $f_i$, $\mu_X$ to $\mathcal{N}(x, \sigma^2 I)$ and $\mu_Y$ to $\mathcal{N}(x', \sigma^2 I)$, and find a $t$, such that, $\mathbb{P}(X \in S) = \underline{p_i}(x)$. Now, since $\mu_X$ and $\mu_Y$ are isometric Gaussians with the same variance,

$$\frac{\mu_Y(z)}{\mu_X(z)} \leq t \iff (x' - x)^T z \leq \beta$$

for some $\beta \in \mathbb{R}$. Therefore, the set $S$ is a half-space defined by a hyper-plane orthogonal to the perturbation $x' - x$. So, if $\|x' - x\|_2 \leq R$, then $\mathbb{P}(Y \in S) \geq \Phi_\sigma(\Phi_\sigma^{-1}(\underline{p_i}(x)) - R)$.

$$
\begin{aligned}
\bar{f}_i(x') &= \mathbb{E}[f_i(Y)] \\
&\geq (b-a)\mathbb{P}(Y \in S) + a && \text{(from the above lemma)} \\
&\geq (b-a)\Phi_\sigma(\Phi_\sigma^{-1}(\underline{p_i}(x)) - R) + a \\
&= b\Phi_\sigma(\Phi_\sigma^{-1}(\underline{p_i}(x)) - R) + a(1 - \Phi_\sigma(\Phi_\sigma^{-1}(\underline{p_i}(x)) - R))
\end{aligned}
$$

The upper bound on $\bar{f}_i(x')$ can be derived similarly by applying the second statement of the above lemma.

### A.1 Alternate proof

Theorem 1 can also be proved for $\sigma = 1$ using Lemma 2 from Salman et al. in [25]. This lemma states that for any function $g : \mathbb{R}^d \to (0, 1)$, $\Phi^{-1}(\bar{g})$ is 1-Lipschitz, where $\Phi^{-1}$ is the inverse CDF of the standard Gaussian distribution. Set $g(.)$ to be $\frac{f_i(.) - a}{b - a}$ for an arbitrary class $i$. Then, $\bar{g}(x) = \frac{\bar{f}_i(x) - a}{b - a}$ is upper and lower bounded by $\overline{p_i}(x)$ and $\underline{p_i}(x)$ respectively. Due to the Lipschitz condition, we have,

$$
\begin{aligned}
\Phi^{-1}(\bar{g}(x)) - \Phi^{-1}(\bar{g}(x')) &\leq \|x - x'\|_2 \leq R \\
\Phi^{-1}(\bar{g}(x')) \geq \Phi^{-1}(\bar{g}(x')) - R &\geq \Phi^{-1}(\underline{p_i}(x)) - R \\
\bar{g}(x') &\geq \Phi(\Phi^{-1}(\underline{p_i}(x)) - R)
\end{aligned}
$$

Substituting $\bar{g}(x) = \frac{\bar{f}_i(x) - a}{b - a}$ and rearranging terms appropriately gives us the first bound in theorem 1. The second bound can be derived similarly.

## B   Proof of Lemma 3

Let $S^c$ be the complement set of $S$.

$$
\begin{aligned}
\mathbb{P}(h(Y) \geq s) - \mathbb{P}(Y \in S) &= \int_{\mathbb{R}^d} \mathbf{1}\{h(z) \geq s\}\mu_Y(z)dz - \int_S \mu_Y(z)dz \\
&= \left[ \int_{S^c} \mathbf{1}\{h(z) \geq s\}\mu_Y(z)dz + \int_S \mathbf{1}\{h(z) \geq s\}\mu_Y(z)dz \right] - \int_S \mu_Y(z)dz \\
&= \int_{S^c} \mathbf{1}\{h(z) \geq s\}\mu_Y(z)dz - \int_S (1 - \mathbf{1}\{h(z) \geq s\})\mu_Y(z)dz \\
&\geq t \left[ \int_{S^c} \mathbf{1}\{h(z) \geq s\}\mu_X(z)dz - \int_S (1 - \mathbf{1}\{h(z) \geq s\})\mu_X(z)dz \right] \\
&\qquad\qquad\qquad\qquad\qquad\qquad\qquad (\text{since } 0 \leq \mathbf{1}\{h(z) \geq s\} \leq 1) \\
&= t \left[ \int_{\mathbb{R}^d} \mathbf{1}\{h(z) \geq s\}\mu_X(z)dz - \int_S \mu_X(z)dz \right] \\
&= t \left[ \mathbb{P}(h(X) \geq s) - \mathbb{P}(X \in S) \right] \geq 0
\end{aligned}
$$

The second statement of the lemma can be proven similarly by switching $\geq$ and $\leq$.

## C   Additional Experiments

In section 5.2, we compared the two methods, using Hoeffding's inequality and Dvoretzky–Kiefer–Wolfowitz inequality to derive the required lower bounds, for the certificates. We

(a) Average Prediction Score (CIFAR-10)

(b) Margin (CIFAR-10)

(c) Average Prediction Score (ImageNet)

(d) Margin (ImageNet)

Figure 1: Certified accuracy vs. radius (CIFAR-10 & ImageNet) at different cutoffs for average confidence score with $\sigma = 0.50$. Solid and dashed lines represent certificates computed with and without CDF bound respectively.

repeat the same experiments in figure 1 for $\sigma = 0.50$. Then, in figure 2, we show that the CDF-based method (using the DKW inequality) outperforms the baseline approach regardless of how tight a lower-bound for $e_i(x)$ is used in the baseline certificate (1). We replace $\underline{e_i}(x)$ with the empirical estimate of the expectation $\hat{e}_i(x) = \sum_{j=1}^{m} f_i(x + \delta_j)/m$, which is an upper bound on $\underline{e_i}(x)$. And since bound (1) is an increasing function of $\underline{e_i}(x)$, any valid lower bound $\underline{e_i}(x)$ on the expectation cannot yield a certified accuracy better than that obtained using $\hat{e}_i(x)$. We compare our certificate with the best-possible baseline certificate for some of Cohen et al.'s ResNet-110 models trained on the CIFAR-10 dataset using the same value of $\alpha$ as in section 5.2. The baseline mostly stays below the CDF-based method for both types of confidence measures under the noise levels considered.

(a) Average Prediction Score at $\sigma = 0.25$

(b) Margin at $\sigma = 0.25$

(c) Average Prediction Score at $\sigma = 0.50$

(d) Margin at $\sigma = 0.50$

Figure 2: Certified accuracy vs. radius (CIFAR-10 only) at different cutoffs for average confidence score. Solid lines represent certificates computed with the CDF bound and dashed lines represent the best-possible baseline certificate.