[Reviews · NeurIPS 2020]

Review 1

Summary and Contributions: This paper essentially claims two contributions: (1) it proposes to do randomized smoothing on soft confidence scores rather than on hard classifications, and (2) it presents a CDF-based robustness certificate (applicable to randomized smoothing on soft scores but not hard labels) that is potentially tighter than the standard expectation-based robustness certificate. I like contribution (2), but I am less excited about contribution (1), because soft smoothing was already discussed in [https://arxiv.org/abs/1906.04584].

Strengths: The paper proposes an interesting randomized smoothing certificate that is based on the distribution of confidence scores rather than on just their expectation. Typically, in randomized smoothing, one is told the expectation, over random Gaussian noise, of f(x+noise), and the goal is to lower-bound the expectation of f(x'+noise), where x' is a nearby point and the output of the function f is assumed to be bounded -- say between 0 and 1. The "worst case" f, which attains the lower bound, is a piecewise constant function with steps orthogonal to the perturbation (x' - x) which puts 1's on the side closer to x and 0's on the side closer to x'. The technical observation in this paper is that we can do better if we are told not just the expectation of f(x+noise) but its CDF. (Like the mean, the CDF can be estimated from samples.). Intuitively, the "worst case" f from the original bound has the distribution of f(x+noise) being very anti-concentrated, so if we are told that, to the contrary, this distribution must be concentrated, then that rules out the original bound's worst case. The new bound proved in this paper is similar to the old bond, in that the "worst case" base classifier is still a piecewise constant function with steps orthogonal to the perturbation (x' - x). For example, we are told that P[ f(x+noise) < 0.4 ] = 0.8 and P[ f(x+noise) < 0.6] = 1.0, then the new "worst case" f is a piecewise constant function which puts 0.6's on the side closer to x and 0.4's on the side closer to x'.

Weaknesses: The weaknesses of this paper mostly have to do with the motivation of confidence smoothing: -- Confidence smoothing was already discussed in Salman et al (https://arxiv.org/abs/1906.04584) under the name "soft smoothing"; your Theorem 1 is an immediate application of their Lemma 2. -- As the authors write, deep network classifiers are not well-calibrated, so it's unclear how valuable it is to have the confidence information. -- The plot in Figure 4 looks to me like it does show a correlation between prediction score and certified radius. The paper would be stronger if you could demonstrate that confidence smoothing with the CDF-based certificate outperforms hard label smoothing with the standard expectation-based certificate. As is, you do show that confidence smoothing with the CDF-based certificate outperforms confidence smoothing with the expectation-based certificate, but that isn't a reason to use confidence smoothing in the first place.

Correctness: I think so.

Clarity: Yes.

Relation to Prior Work: Yes, except the authors do not discuss "soft smoothing" from Salman et al.

Reproducibility: Yes

Additional Feedback: Update: I have read the rebuttal, and I'm keeping my score. I think that the motivation is a little bit weak, given the prior Salman et al. paper, but this submission is possibly over the bar. I just realized that Lemma 2 from Salman et al. only covers the case where \sigma = 1. That said, here is how to use Lemma 2 from Salman et al. to prove your Theorem 1, in the special case where \sigma = 1. Lemma 2 from Salman et al states that if g is any function whose output is bounded in [0, 1] then the function \Phi( \bar{g} ) is Lipschitz with constant 1, which implies: \Phi^{-1}( \bar{g} (x') <= \Phi^{-1}( \bar{g} (x')) + |||x' - x|| Applying the monotone function \Phi to both sides of this inequality yields: \bar{g} (x') <= Phi(\Phi^{-1}( \bar{g} (x')) + |||x' - x||) Now, to prove your Theorem 1 using this result, we define the function g(x) = (f(x)-a)/(b-a), which is bounded in [0, 1] as required. We then apply that lemma, then multiply both sides by (b-a), then add a, and then rearrange, to obtain the first statement of Theorem 1. (The second statement can be proved in the same way.) To generalize to the case where \sigma is not 1, you could invoke Lemma 1 from Levine et al (https://arxiv.org/abs/1905.12105), which is the exact same thing as the Salman et al lemma, but for general \sigma.


Review 2

Summary and Contributions: This paper investigates certified defense regarding confidence scores (rather than mere classification) via randomized smoothing. By considering *confidence distribution*, the resulting method (theory) beats the baseline method only using the confidence average.

Strengths: In my opinion, confidence certification is very important when deploying machine learning models in real-world applications. The theoretical result is non-trivial, behind which the intuition is well explained.

Weaknesses: See additional feedback for several flaws (or suggestions).

Correctness: The claims and method make sense to me.

Clarity: Yes, the paper is well written and easy to read. The background and the proposed method are clearly presented.

Relation to Prior Work: Related work is introduced from the randomized smoothing view. It would be better to also talk about whether there are privious work focusing on confidence score certification.

Reproducibility: Yes

Additional Feedback: ## After reading the feedback The feedback about parameters s1,..., sn is satisfactory. I expect the discussion is added to text with more details, maybe experiments investigating the parameters. *** * The parameters s_1… s_n are not discussed in text. It would be better to talk about how to set these parameters, and how these parameters affect the performance. * 240: “n = 100,000” should be “m = 100,000”. * I would like to suggest to add experiments on ImageNet — only one dataset (CIFAR-10) is not sufficiently convincing. * It would be better to add discussion about related work on certifying confidence.


Review 3

Summary and Contributions: Randomized smoothing generates a certified radius for a classifier’s prediction without conveying any information about how confident an input point should be in the assigned label. This work proposes to address this problem by restoring the confidence information in certified classifiers and further averaging the confidence scores (measured by average prediction score and the margin) from the underlying base classifier for prediction.

Strengths: - The motivation of measuring confidence is clear. - Theorems are carefully proved. - The confidence measure helps the performance.

Weaknesses: - Some model design rationale is unclear. - Experimental studies relying on single dataset may not be convinced. - It lacks detailed experimental analysis.

Correctness: The authors provide a working and correct solution for the problem, which seems technical sound.

Clarity: The presentation is good and clear.

Relation to Prior Work: The paper is missing a section discussing related works regarding randomized smoothing. It’s strongly suggested to include a detailed discussion how the contribution made by this paper differs from prior works.

Reproducibility: Yes

Additional Feedback: - In page 3, how to determine the parameter a, b and c? - The paper uses two notions to measure confidence, i.e., the average prediction score, and the margin. The definition of these two notions are straightforward; the rationale of such design rational is not explicit and sound – i.e., why these two notions can be used for confidence measure. - The current experiments are conducted using ResNet-110 on CIFAR-10. It may not be sufficient. The results on ImageNet are also recommended as Cohen et al. [6] did. - What is the relationship between radii \sigma and confidence threshold? Why the authors select different thresholds for \sigma=0.25 and 0.5 in Figure 2?


Review 4

Summary and Contributions: The authors extend the idea of providing certified radii around input points for class assignment to the provision of confidence. This deals with the issue that, just because it is confidently not an adversarial example doesn't mean it's not very uncertainty about the actual class. They use the underlying base classifier's confidence scores at the samples and use this to determine a bound on the capacity for an adversary to manipulate it.

Strengths: Uncertainty quantification is often forgotten in ML work. This seems a useful paper which allows users of such tools to also have access to the confidence of the classifier - this is something a user might use to make decisions with and so it is valid that it also should be protected from adversarial attack. I am not aware of work quite on this topic & so it has novelty. The claims (and theorems) appear to be sound. I like the empirical experiments - in particular the check to see what relationship exists between the softmax confidence score and the radius.

Weaknesses: I can't see serious weaknesses. Maybe application to another dataset? Also could this be applied to other classifiers (not DNNs?)

Correctness: I'm not very confident - but I think they are.

Clarity: I also like the number of examples used throughout. It is clear prose. The mathematically lemmas are accompanied by coherent proofs.

Relation to Prior Work: It seems to discuss the field sufficiently to place itself as a novel contribution. There is some work by Kathrin Grosse (e.g. "The Limitations of Model Uncertainty in Adversarial Settings") that concerns itself with uncertainty I think?

Reproducibility: Yes

Additional Feedback: line 80: typo "withing" -> with in it might be nice for the figure numbers to be in the order of the text? line 193: The equations surely should be mu_X and mu_Y in the ratios. ===Having read the other reviews and feedback=== I have low confidence in this area. However, for me the two main issues are now: 1) The need for an additional experiment (as suggested by the other reviewers). 2) There seems to be literature that has been missed by the authors (e.g. that mentioned by reviewers 1 & 3. including Salman et al...) However: I feel that these can be added as requirements for the changes after submission, for the camera ready version however, so I'm happy to leave my score the same.

[Author Response · NeurIPS 2020]

**ImageNet results (R2, R3 & R4):** As suggested by most reviewers, we perform our experiments on the ImageNet dataset by certifying a pre-trained ResNet-50 from Cohen et al. in [6] and produce the following plots similar to Figure 2 in our paper. Just like the CIFAR-10 results, the CDF-based method significantly outperforms the naive baseline for different levels of smoothing noise $\sigma$. Due to space constraints, we show plots for $\sigma = 0.25$ noting that plots for $\sigma = 0.5$ demonstrate similar trends.

**Calibration of confidence information (R1):** We agree that it is improper to interpret confidence scores as the "probability" of the classification being correct – indeed, this is the reason why we prefer the term "class score" over "class probability." However, we think it is widely accepted that scores are often strongly correlated with the chance of correct classification, even if this is not an exact or linear relationship because of poor calibration.

**Correlation between prediction score and certified radius (R1):** We do not mean to imply no correlation exists, but rather that the correlation in Fig 4 is *weak*; there are many images with large radius but low confidence, and visa versa.

**Theorem 1 is an immediate application of Lemma 2 in Salman et al. (R1):** While the two theorems might be related, the connection does not seem so obvious for us. Theorem 1 in our paper is regarding the *baseline* certificate against which we compare our CDF-based certificate (Theorem 2) which is our main technical contribution. We will make a reference to this lemma in the paper.

(a) Average Prediction Score

(b) Margin

Figure 1: ImageNet Results ($\sigma = 0.25$).

**Parameters $s_1, \ldots, s_n$ are not discussed in text (R2):** In our experiments, we set $s_1, \ldots, s_n$ such that the number of confidence score values in every interval $(a, s_1), (s_1, s_2), \ldots, (s_n, b)$ is equal. We chose this approach instead of setting the parameters at regular intervals in $(a, b)$ because doing so would make a lot of intervals empty. It made sense to split the range in such a way that the intervals are well-balanced. There could be other reasonable ways to set these parameters which could have an impact on the performance. We will add this discussion to the paper.

**In page 3, how to determine the parameter a, b and c? (R3):** Parameters $a$ and $b$ depend on the confidence measure used and denote the range $(a, b)$ in which the confidence scores lie. For instance, if you are using average prediction score, then $a = 0$ and $b = 1$ (see page 6 under section 4: Confidence measures) and if you are considering margin of average prediction score, then $a = -1$ and $b = 1$. Parameter $c$ is the confidence threshold you would like to certify, i.e., the confidence score is guaranteed to be above $c$ within the certified radius. This parameter can be set by the user to calculate the corresponding certified radius. In our experiments, to keep computing requirements within reach, we compute the threshold $c$ for a chosen radius instead. But, the original task of computing the radius for a given $c$ can be accomplished using a simple binary-search. (See page 5, lines 172-175).

**The rationale behind confidence notions (R3):.** The *average* score is a very conventional measure of classification confidence, and is the more direct analog of classical confidence scores in the setting of a smoothed classifier. The classifier *margin* is less conventional; it measures how much more certain the top class is over the second-place class. We find that the distribution of margins is often more concentrated than average scores, and we can therefore produce certificates with stronger bounds. There is a tradeoff here that the user can choose – the average is more interpretable but provides weaker bounds, the margin is less conventional (although large margin is still a good indicator of confidence) and provides tighter certificates.

**Relationship between radii $\sigma$ and confidence threshold (R3):** If you consider two Guassians separated by $\epsilon$ units, their overlap will be large when $\sigma$ is large relative to $\epsilon$, and the overlap is very small when $\sigma$ is small. For this reason, when $\sigma$ is large, the confidence bound decreases more slowly with increasing $\epsilon$, allowing us to certify larger radii. However, when $\epsilon$ is small (e.g., for certifying small radii), the overlap between Gaussians is large, even for small $\sigma$, and so we can produce certificates with small $\sigma$. In this case, we can take advantage of the fact that the sampled scores are more homogenous when $\sigma$ is small to get tighter bounds. Our experiments show numerous values of $\sigma$ to show that our results hold for a range of choices. However in practice the optimal $\sigma$ is proportional to the radius you want to certify. We will include this discussion when we update our paper.

**Related work (R2 & R3):** We will be sure to add a section on related work in randomized smoothing and certifying confidences.

[Meta-Review · NeurIPS 2020]

Thank you for your submission to NeurIPS. While some of the reviewers felt that some aspects of this paper lacked much novelty (particularly, there was some debate about whether the technical results could be derived easily from existing results in the literature), there was also a consensus that the broad technique of using the full CDF of classifier scores in order to tighten randomized smoothing certification radii was an interesting one, and worthy of publication. I would recommend that in revising their paper, the authors strongly look at the connections to previous theory as mentioned by the reviewers, and determine whether the proofs they present can be more easily derived as consequences of past results (even as alternative derivations, if they feel the "self-contained" derivation is preferable to include). However, this doesn't diminish the overall value of the paper, which I still believe to be strong enough to warrant acceptance.